# Causal speech enhancement using dynamical-weighted loss and attention encoder-decoder recurrent neural network

**Fahad Khalil Peracha[1], Muhammad Irfan Khattak[1], Nema Salem[2]\*, Nasir Saleem[3]**

**1** Department of Electrical Engineering, University of Engineering and Technology, Peshawar, KPK, Pakistan, **2** Electrical and Computer Engineering Department, Effat College of Engineering, Effat University, Jeddah, KSA, **3** Department of Electrical Engineering, University of Engineering and Technology, Peshawar, KPK, Pakistan

\* nasirsaleem@gu.edu.pk

**Data Availability Statement:** All relevant data are within the manuscript and its Supporting information files.

## Abstract

Speech enhancement (SE) reduces background noise signals in target speech and is applied at the front end in various real-world applications, including robust ASRs and real-time processing in mobile phone communications. SE systems are commonly integrated into mobile phones to increase quality and intelligibility. As a result, a low-latency system is required to operate in real-world applications. On the other hand, these systems need efficient optimization. This research focuses on the single-microphone SE operating in real-time systems with better optimization. We propose a causal data-driven model that uses attention encoder-decoder long short-term memory (LSTM) to estimate the time-frequency mask from a noisy speech in order to make a clean speech for real-time applications that need low-latency causal processing. The encoder-decoder LSTM and a causal attention mechanism are used in the proposed model. Furthermore, a dynamical-weighted (DW) loss function is proposed to improve model learning by varying the weight loss values. Experiments demonstrated that the proposed model consistently improves voice quality, intelligibility, and noise suppression. In the causal processing mode, the LSTM-based estimated suppression time-frequency mask outperforms the baseline model for unseen noise types. The proposed SE improved the STOI by 2.64% (baseline LSTM-IRM), 6.6% (LSTM-KF), 4.18% (DeepXi-KF), and 3.58% (DeepResGRU-KF). In addition, we examine word error rates (WERs) using Google's Automatic Speech Recognition (ASR). The ASR results show that error rates decreased from 46.33% (noisy signals) to 13.11% (proposed) 15.73% (LSTM), and 14.97% (LSTM-KF).

## 1 Introduction

Speech signals in many real-world situations are degraded by noise signals. A degraded signal severely influences the performance of many speech-related applications, such as automatic speech recognition [1], speaker identification [2], and hearing aid devices [3]. A speech

**Funding:** Enter: The author(s) received no specific funding for this work.

**Competing interests:** Enter: The authors have declared that no competing interests exist.

enhancement system is mainly involved in restoring the quality and improving the intelligibility of the signals degraded by the noise and is used at the front end of many speech applications to enhance their performance in noisy situations where they are altered. Background noises, competing speakers, and room reverberation are mainly the major sources of variations and distortions. An SE algorithm ideally ought to work passably in various acoustic situations, a broad-spectrum algorithm that is capable of performing well with little complexity and latency in every noisy situation is a challenging technical task.

The conventional approaches include spectral subtraction [4], Wiener filtering [5], statistical models [6–8], and hybrid SE models [9, 10] They show better performance in many stationary noises but face difficulties in handling nonstationary noises. In the recent past, deep learning has been developed into the mainstream for speech enhancement [11]. Given a speech dataset of the clean-noisy pairs, the neural networks can learn to transform the noisy magnitude spectra to their clean counterparts (mapping based) [12–14] or estimate the time-frequency masks (masking-based) such as ideal binary mask (IBM) [15, 16], ideal ratio mask (IRM) [17, 18], and spectral magnitude mask (SMM) [19]. Fully connected networks (FCN) [19], feedforward neural networks (FDNN) with Kalman filtering [20], recurrent neural networks (RNN) [21–23], and convolutional neural networks (CNN) [24, 25] are important deep learning approaches in SE.

A fully connected feedforward neural network showed that a DNN trained for a large number of background noises with a single speaker generalized better to untrained noise types [26]. Such a network, however, shows the difficulty in generalizing to both untrained speakers and noises when trained with a large number of speakers and noises. The RNN with LSTM is used to design a noise- and speaker-independent network for speech enhancement. A four-layered RNN network was used to train speech utterances belonging to 77 different speakers combined with 10,000 different noise types [26]. Recently, SE has aimed to improve the performance of the speaker and noise-independent networks. In [25], a CNN with gated and dilated convolutions is proposed for the magnitude enhancement. A recent trend is the use of attention mechanisms to improve the quality and intelligibility of noisy speech signals. In [27] a speech enhancement approach is proposed and used with an attention LSTM by replacing the forgetting gate with an attention gate. In [24] a dense CNN with a self-attention is proposed to assist feature extraction using a combination of feature reuse. In [28] a dual-path self-attention RNN is proposed to improve the long sequence of speech frames. A number of deep learning studies based on the attention mechanism for SE are successfully proposed with novel results [29–34].

In most of the deep learning approaches, mean-square error (MSE) is used as the loss function [15–19]. Other loss functions include Huber and mean absolute error (MAE). The gradient of the MAE remains invariant during training when loss approaches zero, resulting in missing the minima. Moreover, the Huber loss needs hyperparameter tuning. This brings further complexity when a loss function is dynamically weighted. A large error indicates poor learning on a particular instance in the dataset. A dynamically weighted loss function is used to alter the learning process by augmenting the weighted values corresponding to the learning errors. Through such an amendment, the loss function focuses on the large learning errors and improves the network performance.

In this paper, an attention encoder-decoder LSTM network for sequence-to-sequence learning is proposed. The motivation behind this research is the recent success of the attention mechanism in speech emotion recognition [33] and speech recognition [34]. Deep learning approaches can be regression or prediction tasks [35, 36]. It is useful to employ the attention process in speech enhancement since a human can focus on a certain part of a speech stream with more attention, such as target speech, whereas they perceive the surrounding noise with

less attention. We have used an attention process on the encoder-decoder LSTM network that has been shown to perform better in modeling vital sequential information. LSTM [37–39] can learn the weights of the past input features perfectly and predict the enhanced frames. The attention process determines the correlations between the previous frames and the current frames be enhanced and assigns weights to the previous frames. Experiments have shown that the proposed network consistently performed better in terms of speech quality and intelligibility. The overall structure of the proposed speech enhancement algorithm is depicted in Fig 1. We have summarized the main contributions of this study.

- For sequential learning to handle real-time speech applications that need low-latency causal processing, a causal speech enhancement based on attention encoder-decoder LSTM network is proposed.

- By adding weighted values for large learning errors, a dynamically weighted loss function is used to improve the learning process. The loss function focuses on the large learning errors to further improve the network performance.

- Automatic speech recognition is evaluated using estimated magnitude, thereby notably improving the word error rate in noisy situations.

The remainder of this paper is organized as follows. In Section 2, we explain the proposed speech enhancement algorithm. The dynamically-weighted loss is presented in Section 3. The experimentation is presented in Section 4. The results and discussions are presented in Section 5. Finally, the conclusions are drawn in Section 6.

## 2 Proposed speech enhancement

For a given clean speech signal $x_t$ and noise signal $d_t$, the noisy speech signal $y_t$ is formed by the additive mixing as follows:

$$y_t = x_t + d_t \tag{1}$$

where $\{x, y, d\} \in \mathbb{R}^{N \times 1}$ and $N$ shows number of the speech frames. A SE algorithm aims to recover a close estimate $\hat{x}_t$ of the clean speech $x_t$ given $y_t$. The inputs to the LSTM Encoder-Decoder are $\mathbf{Y} = [y_1, .., y_t, .., y_N]$, where $y_t$ indicates the spectral magnitudes of the noisy speech at frame $t$. The high-level features $h$ are extracted by the encoder from the input speech frames:

$$h^K, h^Q = Encoder(y_t) \tag{2}$$

where $h^K$ and $h^Q$ stand for the key and query, respectively. In this study, unidirectional LSTM is used as an encoder which shows a strong ability to model the sequential data leading to the improved performance of the speech enhancement [39]. The attention process is fed with key and query as the input to create fixed-length context vectors:

$$C^t = Attention(h^K, h^Q) \tag{3}$$

The decoder output $w_t$ is the recovered enhanced speech signal $\hat{x}_t$ which takes the context vectors $C^t$, the output of the encoder $h^Q$, and the noisy speech $y_t$, respectively.

$$w(t) = decoder(C^t, h_t^Q, y_t) \tag{4}$$

The proposed attention encoder-decoder LSTM is depicted in Fig 2.

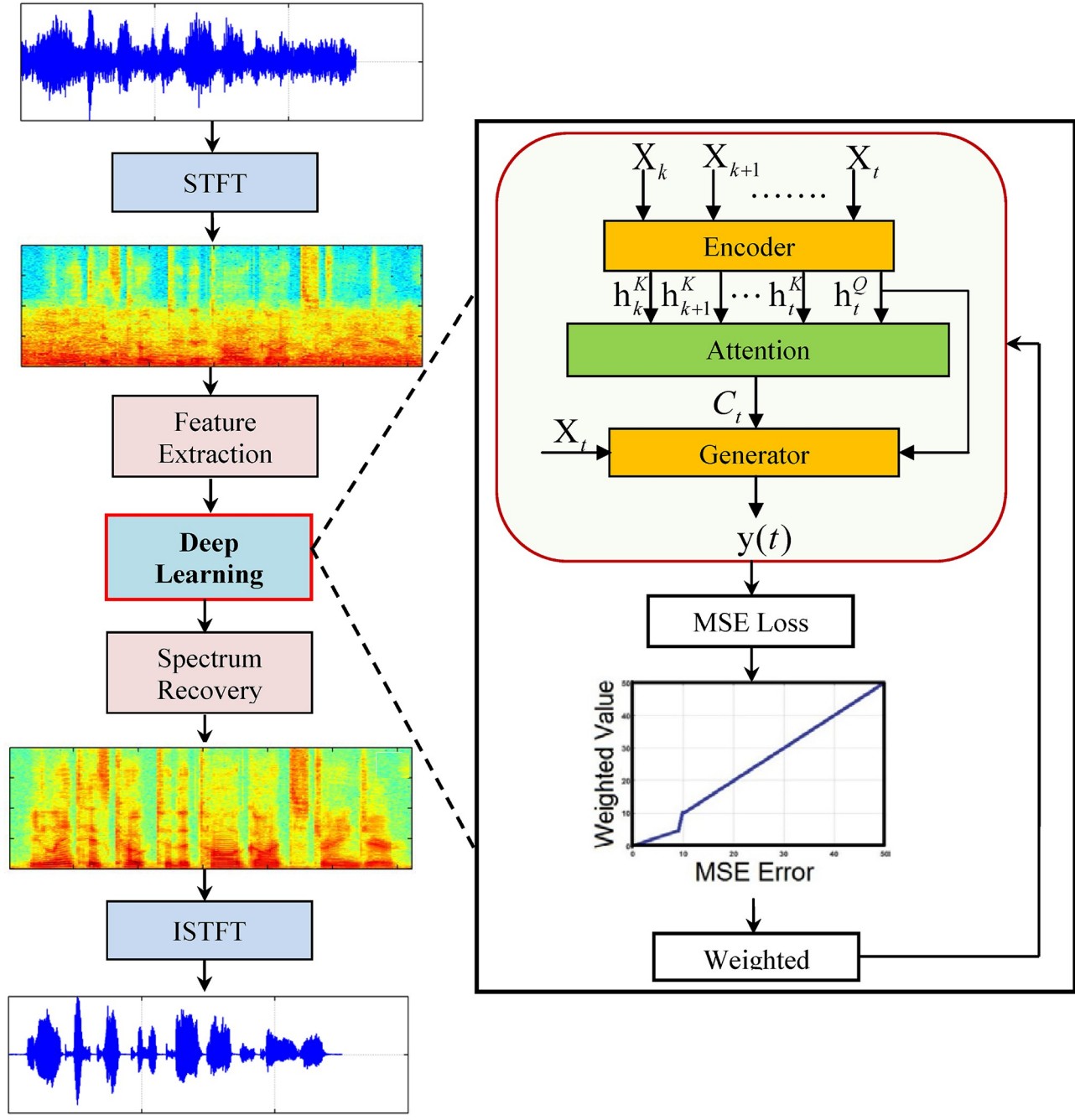

**Fig 1.**

## 2.1 Unidirectional LSTM encoder

The LSTM encoder extracts the high-level feature representations from the input speech frames. The input features are first fed into a fully-connected layer. The $y_t$ is the input to the LSTM cell as:

$$h_t^K = f(y_t) \tag{5}$$

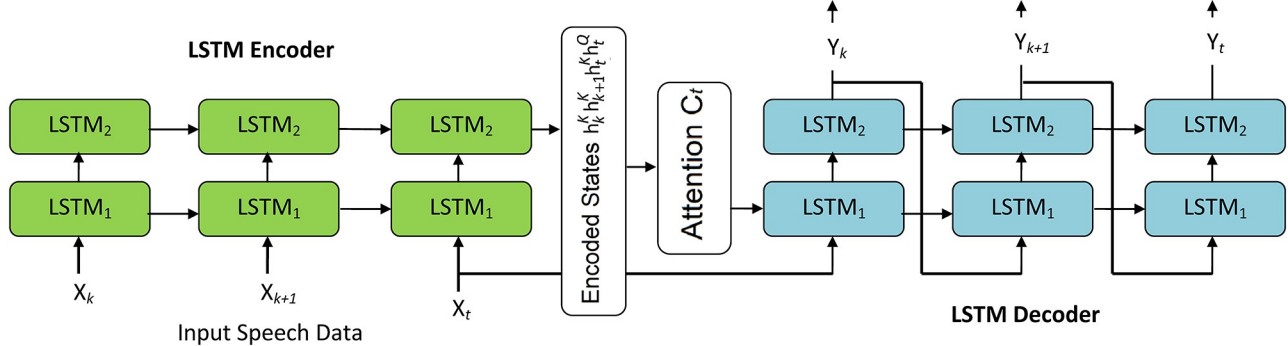

**Fig 2.**

where $f(\cdot)$ is LSTM function whereas $h_t^K$ is LSTM output, respectively. The $h_t^Q$ can be computed as:

$$h_t^Q = f(h_t^K) \tag{6}$$

## 2.2 Attention process

The attention process is fed with information about the key and query as inputs to create fixed-length context vectors. An attention process can use both previous and future speech frames. But, SE is a causal problem and uses previous speech frames to avoid processing latency. We have used casual dynamic and causal local attention approaches. To enhance a speech frame in causal dynamic attention, $\mathbf{Y} = [y_1, .., y_t]$ is used to compute the attention weights which means that all the previous speech frames are used to enhance the current frames. If the duration of the speech utterance is long, the attention weights of several previous speech frames can nearly be zero. Therefore, in casual local attention process, $\mathbf{Y} = [y_1, .., y_t]$ is used to compute the attention weights. The $z$ is set to a constant. The normalized attention weight $\kappa$ can be learned as:

$$\kappa_{tk} = \frac{exp(h_k^K, h_t^Q)}{\sum_{k=l}^{t} exp(h_k^K, h_t^Q)} \tag{7}$$

$l = 1$ for causal dynamic attention whereas $l = (t - z)$ is used for the causal local attention. According to correlation computation, we have:

$$exp(h_k^K, h_t^Q) = h_k^{KT} W h_t^Q \tag{8}$$

The context vector with attention weights is given as:

$$C^t = \sum_{n=l}^{t} (\kappa_t k h_k^K) \tag{9}$$

With an attention-weighted context vector, the model decides the attention process.

**2.2.1 Unidirectional LSTM decoder.** The decoder recovers the output-enhanced speech by using the input features, encoder output, and context vector, respectively. The enhanced vector $E_t$ is learned from context vectors and features as:

$$E_t = tanh(W_E[C^t; h_t^Q] + b_E) \tag{10}$$

where $[C^t; h_t^Q]$ shows the concatenation of the context and feature vectors, respectively. The ideal ratio mask (IRM) is finally estimated from the feature vectors. The time-frequency IRM ($f$, $t$) is given as:

$$IRM(f, t) = \sqrt{\frac{|X(f, t)|^2}{|X(f, t)|^2 + |D(f, t)|^2}} \tag{11}$$

where $|X(f,t)|$ and $|D(f,t)|$ show the magnitude spectra of clean speech and noise signals, respectively. The enhanced vectors are multiplied with the noisy features to recover the enhanced speech by taking the inverse Short-time Fourier Transform (STFT) as:

$$\hat{x}_t = y_t \otimes IRM(f, t) \tag{12}$$

## 2.3 Weighted loss function

In masking-based deep learning methods for SE, a loss function presents a divergence between the predefined and the estimated mask. A loss function aims to reduce the errors produced during training. Mostly, the MSE (mean square error) is used as the basic loss function, given as:

$$loss(m, n) = \frac{1}{L}\sum_{j=0}^{L}((m - n))^2 \tag{13}$$

where $m$, and $n$ denote the input and the predicted value, respectively. Eq (13) can be represented in terms of the time-frequency mask as:

$$loss(M_x, \hat{M}_x) = \frac{1}{L}\sum_{j=0}^{L}((M_x - \hat{M}_x))^2 \tag{14}$$

where $\hat{M}(x)$ and $M(x)$ denote the estimated and predefined IRM masks, respectively. The dynamical-weight loss function is used to adjust the network learning by multiplying weighted values corresponding to the learning errors. Thus, the loss function focuses on large errors to improve performance. The MSE loss function is multiplied by a weighted variable $O$ to get the weighted MSE as:

$$loss(m, n) = O*\left(\frac{1}{L}\sum_{j=0}^{L}((M_x - \hat{M}_x))^2\right) \tag{15}$$

To emphasize the instances with large errors, the weight variable $O$ in Eq (15) is updated according. The weight selection is done according to the following condition:

$$O*Loss(M_x, \hat{M}_x) = \begin{cases} \dfrac{|M_x - \hat{M}_x|}{2}, where |M_x - \hat{M}_x| < B \\[2ex] |(M_x - \hat{M}_x)|, where |M_x - \hat{M}_x| \geq B \end{cases} \tag{16}$$

Where $|.|$ indicates the magnitude of ground truth and estimated masks. The weighting became halved when the absolute divergence is less than constant $B$ which is set to 10 since it has been observed that the performance of the model at this instance was better.

## 3 Experiments

### 3.1 Data generation

In experiments, we have used IEEE dataset [40]. Two IEEE datasets are used which are composed of male speakers and female speakers, respectively. The noise sources are selected from AURORA [41]. For a given speech dataset $D$, we have $M_{tr}$ and $M_{te}$ as the training and testing speech utterances. The training and testing speech utterances in the dataset are denoted by $D_{tr}$ and $D_{te}$, respectively. The noisy utterances are generated by adding the noise signals to $D_{tr}$ and $D_{te}$:

$$y_{tr}^i = x_{tr}^i + d_{tr}^i, i = 1, 2, 3, ..., M^{tr} \tag{17}$$

$$y_{te}^j = x_{te}^j + d_{te}^j, j = 1, 2, 3, ..., M^{tr} \tag{18}$$

### 3.2 Feature extraction

The input pairs $y$, $x$, $d$ are transformed from the time to frequency domain using the STFT as:

$$\mathbf{Y} = STFT(y); \mathbf{X} = STFT(x); \mathbf{D} = STFT(d) \tag{19}$$

Where $\left\{ X, Y, D \right\} \in \mathbb{Z}^{T \times F}$, $T$ and $F$ show frame number and frequency bin number. We have used STFT magnitude $|Y|$ as the input features.

### 3.3 Experimental setup

We have used speech utterances with a 16 kHz sampling rate. A 512 points Hanning window with 75% overlapping is used. We used the noisy phase during waveform reconstruction. The network consists of an input layer, three unidirectional LSTMs with 256 memory units followed by a fully connected output layer with 257 sigmoidal units. The number of epochs and the learning rate is set to 160 and 0.001, respectively. The weights are randomly initialized and trained with 32 sequences mini-batches by back-propagation through time with Adam optimizer. The three-layered LSTM network architecture with (128/256/256/256/257) memory cells is used. The details of hyperparameters are given in Table 1. To create the noisy utterances, three SNR levels are used (-5dB to +5dB) with a 5dB step size. For network training, IEEE speech utterances from the male and female speakers are duplicated three times for each SNR level and mixed with all noise types. Therefore, a total of 21600 (approx. 18 hours) speech utterances are used in the training process. We also used half-speech utterances during testing in the matched and mismatched conditions. During testing, each noise type is tested with a different set of utterances.

To evaluate the performance of the proposed approach, we used three objective measures: Short-time objective intelligibility (STOI) [42], Perceptual evaluation of speech quality (PESQ) [43], and Source-to-distortion ratio (SDR). To compare performance of the proposed method, baseline LSTM [26], DNN [18], LMMSE [6], OM-LSA [7], FDNN-KF [20], LSTM-KF [44] DeepXi-KF [45] and DeepResGRU-KF [46]. are selected as the competing methods. The competing deep learning methods can be represented as LSTM-IRM: LSTM with IRM as a training target. DNN-IRM: Feedforward DNN with IRM as a training target. DWAtten-LSTM-IRM: Proposed method with IRM as a training target.

**Table 1. Hyperparameters of all deep learning networks.**

| Hyperparameters | Atten-LSTM | LSTM | LSTM-KF | DNN |
|---|---|---|---|---|
| No of layers | 3 | 3 | 3 | 3 |
| Layer 1 Neurons | 256 | 256 | 256 | 1024 |
| Layer 2 Neurons | 256 | 256 | 256 | 1024 |
| Layer 3 Neurons | 256 | 256 | 256 | 1024 |
| Learning Rate | 0.0001 | 0.0001 | 0.0001 | 0.0001 |
| No of Epochs | 160 | 160 | 160 | 160 |
| Momentum | 0.9 | 0.9 | 0.9 | 0.9 |

## 4 Results and discussions

To signify the performance of the proposed speech enhancement method, we have compared the results with baseline LSTM-IRM [26], DNN-IRM [18], LMMSE [6], OM-LSA [7], FDNN-KF [20], LSTM-KF [44] DeepXi-KF [45] and DeepResGRU-KF [46]. For matched and unmatched conditions, the average STOI, PESQ, and SDR test values across several noise sources and SNRs are given in Tables 2–5. Note that, in contrast to the competing deep-learning methods, the proposed DWAtten-LSTM presents the highest performance in terms of the STOI, PESQ, and SDR values in noisy situations. Two conditions including Matched and Mismatched are considered in experiments. During match conditions, the speakers and utterances remain the same in training and testing sets whereas, in mismatched conditions, the speakers and utterances in the training set are different from the testing set.

In the matched conditions (Tables 2–4), the proposed DWAtten-LSTM approach achieved the best STOI, PESQ, and SDR values with airport noise, street noise, and car noise at SNR$\geq$5dB, that is, STOI$\geq$94%, PESQ$\geq$2.99, and SDR$\geq$10.8dB. The STOI with the babble noise is improved from 68.1% with noisy speech signals to 85.2% with DWAtten-LSTM and achieves 17.1% improvement in STOI at 0dB SNR. Similarly, the PESQ with airport noise is improved from 1.93 with DNN-IRM to 2.29 with proposed DWAtten-LSTM and improved the PESQ by factor 0.36 (18.75%) at -5dB SNR. Moreover, the SDR value with car noise is improved from 4.20dB with LSTM-IRM to 4.56dB with DWAtten-LSTM and achieved 0.36dB (8.57%) improvement at -5dB SNR level. Note from Table 2 (matched condition) that DWAtten-LSTM presents the highest STOI, PESQ, and SDR values in all noisy situations as compared to baseline LSTM with the same network architecture.

The average STOI, PESQ, and SDR test values across all noise sources and SNRs are given in Table 5 for unmatched conditions where the proposed DWAtten-LSTM achieved the best STOI, PESQ, and SDR values at airport noise at SNR$\geq$5dB, that is, STOI$\geq$92.3%, PESQ$\geq$2.86, and SDR$\geq$10.7dB. The STOI with factory noise is improved from 55.2% with noisy speech signals to 77.0% with DWAtten-LSTM and achieves 21.8% STOI gain at -5dB SNR. Fig 3 shows STOI, PESQ, and SDR improvements in various background noises where we can see the performance of the proposed DWAtten-LSTM in individual noise at -5dB, 0dB, and 5dB SNRs.

We also compared DWAtten-LSTM to non-deep learning-based LMMSE and OM-LSA. Table 6 shows STOI, PESQ, and SDR values obtained with DWAtten-LSTM, LMMSE, and OM-LSA, respectively. STOI is improved from 70.22% and 71.50% with LLMSE and OM-LSA to 85.60% with DWAtten-LSTM and achieves 15.38% and 14.10% STOI gain at 0dB. Similarly, PESQ is improved from 2.22 and 2.20 with LMMSE, and OM-LSA to 2.85 with proposed DWAtten-LSTM and achieved 0.63 and 0.65 PESQ gain at 5dB. The PESQi and STOIi are demonstrated in Fig 4.

**Table 2. STOI (in%) in matched test scores.**

| Noise | Algorithm | -5dB | 0dB | 5dB | Avg |
|---|---|---|---|---|---|
| Airport Noise | Noisy (UnP) | 62.4 | 73.7 | 83.9 | 73.3 |
| | DNN-IRM | 80.2 | 86.5 | 90.7 | 85.8 |
| | LSTM-IRM | 82.9 | 88.6 | 92.3 | 87.9 |
| | LSTM-KF | 74.2 | 80.3 | 82.8 | 79.1 |
| | FDNN-KF | 72.2 | 79.8 | 81.3 | 77.8 |
| | DeepXi-KF | 75.9 | 81.2 | 84.7 | 80.6 |
| | DeepResGRU-KF | 76.6 | 82.3 | 85.5 | 81.1 |
| | Proposed | 85.6 | 90.5 | 94.0 | 90.0 |
| Babble Noise | Noisy (UnP) | 56.7 | 68.1 | 79.6 | 68.1 |
| | DNN-IRM | 72.0 | 79.3 | 85.9 | 79.0 |
| | LSTM-IRM | 75.9 | 82.6 | 87.0 | 81.8 |
| | LSTM-KF | 74.1 | 80.1 | 82.7 | 79.2 |
| | FDNN-KF | 72.3 | 79.6 | 81.1 | 78.0 |
| | DeepXi-KF | 75.8 | 81.3 | 84.8 | 80.7 |
| | DeepResGRU-KF | 76.5 | 82.4 | 85.6 | 81.2 |
| | Proposed | 79.8 | 85.2 | 89.2 | 84.7 |
| Car Noise | Noisy (UnP) | 58.8 | 68.9 | 79.6 | 69.1 |
| | DNN-IRM | 73.7 | 81.0 | 86.0 | 80.2 |
| | LSTM-IRM | 78.5 | 84.7 | 89.1 | 84.1 |
| | LSTM-KF | 78.3 | 83.2 | 87.6 | 83.0 |
| | FDNN-KF | 75.4 | 80.4 | 83.3 | 79.7 |
| | DeepXi-KF | 80.1 | 85.6 | 89.0 | 84.9 |
| | DeepResGRU-KF | 82.2 | 86.8 | 91.2 | 86.7 |
| | Proposed | 83.4 | 88.5 | 92.6 | 88.2 |
| Factory Noise | Noisy (UnP) | 56.4 | 67.4 | 78.8 | 67.5 |
| | DNN-IRM | 70.2 | 78.7 | 85.2 | 78.0 |
| | LSTM-IRM | 74.4 | 80.2 | 88.1 | 80.9 |
| | LSTM-KF | 74.1 | 78.5 | 83.6 | 78.7 |
| | FDNN-KF | 70.7 | 76.2 | 81.4 | 76.1 |
| | DeepXi-KF | 77.2 | 79.8 | 85.4 | 80.8 |
| | DeepResGRU-KF | 79.1 | 81.2 | 86.5 | 82.2 |
| | Proposed | 78.6 | 82.1 | 90.0 | 83.6 |
| Street Noise | Noisy (UnP) | 63.0 | 74.0 | 84.0 | 73.7 |
| | DNN-IRM | 78.4 | 85.3 | 90.3 | 84.6 |
| | LSTM-IRM | 82.0 | 86.8 | 91.8 | 87.9 |
| | LSTM-KF | 74.8 | 82.9 | 86.6 | 83.0 |
| | FDNN-KF | 70.8 | 79.8 | 83.3 | 79.7 |
| | DeepXi-KF | 78.3 | 84.9 | 89.0 | 84.9 |
| | DeepResGRU-KF | 80.1 | 85.6 | 91.2 | 86.7 |
| | Proposed | 84.9 | 89.7 | 93.4 | 89.3 |

The overall average STOI, PESQ, and SDR values for all noise sources are given in Table 6 for matched (denoted as Proposed-M), unmatched (denoted by Proposed-UM) conditions, and the average of both matched and unmatched (denoted as Proposed-Avg). PESQ and STOI values are calculated for the causal local attention process, and the value of $z$ is varied from 4 to 12 with an increment of 4. Table 7 shows the results. It is noticed that values greater than 12

**Table 3. PESQ in matched test scores.**

| Noise | Algorithm | -5dB | 0dB | 5dB | Avg |
|---|---|---|---|---|---|
| | Noisy (UnP) | 1.53 | 1.86 | 2.14 | 1.84 |
| Airport Noise | DNN-IRM | 1.93 | 2.41 | 2.79 | 2.37 |
| | LSTM-IRM | 2.11 | 2.54 | 2.87 | 2.51 |
| | LSTM-KF | 1.88 | 2.13 | 2.56 | 2.14 |
| | FDNN-KF | 1.79 | 2.03 | 2.45 | 2.07 |
| | DeepXi-KF | 1.99 | 2.21 | 2.58 | 2.23 |
| | DeepResGRU-KF | 2.06 | 2.33 | 2.71 | 2.34 |
| | Proposed | 2.29 | 2.67 | 2.98 | 2.65 |
| Babble Noise | Noisy (UnP) | 1.52 | 1.75 | 2.07 | 1.78 |
| | DNN-IRM | 1.95 | 2.34 | 2.66 | 2.32 |
| | LSTM-IRM | 2.07 | 2.43 | 2.70 | 2.40 |
| | LSTM-KF | 1.93 | 2.18 | 2.56 | 2.19 |
| | FDNN-KF | 1.84 | 2.13 | 2.45 | 2.12 |
| | DeepXi-KF | 2.04 | 2.26 | 2.60 | 2.28 |
| | DeepResGRU-KF | 2.11 | 2.38 | 2.71 | 2.40 |
| | Proposed | 2.19 | 2.56 | 2.78 | 2.51 |
| Car Noise | Noisy (UnP) | 1.37 | 1.62 | 1.92 | 1.64 |
| | DNN-IRM | 1.76 | 2.17 | 2.58 | 2.17 |
| | LSTM-IRM | 2.01 | 2.41 | 2.78 | 2.40 |
| | LSTM-KF | 2.03 | 2.33 | 2.67 | 2.41 |
| | FDNN-KF | 1.94 | 2.25 | 2.60 | 2.33 |
| | DeepXi-KF | 2.08 | 2.45 | 2.73 | 2.48 |
| | DeepResGRU-KF | 2.21 | 2.55 | 2.86 | 2.52 |
| | Proposed | 2.27 | 2.65 | 2.99 | 2.63 |
| Factory Noise | Noisy (UnP) | 1.31 | 1.61 | 1.92 | 1.61 |
| | DNN-IRM | 1.72 | 2.16 | 2.60 | 2.16 |
| | LSTM-IRM | 1.94 | 2.32 | 2.71 | 2.32 |
| | LSTM-KF | 1.81 | 2.05 | 2.36 | 2.07 |
| | FDNN-KF | 1.72 | 2.01 | 2.28 | 2.01 |
| | DeepXi-KF | 1.92 | 2.12 | 2.40 | 2.15 |
| | DeepResGRU-KF | 1.98 | 2.26 | 2.55 | 2.26 |
| | Proposed | 2.16 | 2.57 | 2.87 | 2.53 |
| Street Noise | Noisy (UnP) | 1.47 | 1.86 | 2.01 | 1.78 |
| | DNN-IRM | 1.94 | 2.44 | 2.65 | 2.34 |
| | LSTM-IRM | 2.11 | 2.53 | 2.80 | 2.48 |
| | LSTM-KF | 1.84 | 2.12 | 2.39 | 2.12 |
| | FDNN-KF | 1.75 | 2.04 | 2.31 | 2.03 |
| | DeepXi-KF | 1.93 | 2.21 | 2.45 | 2.20 |
| | DeepResGRU-KF | 2.01 | 2.31 | 2.65 | 2.32 |
| | Proposed | 2.29 | 2.66 | 2.99 | 2.65 |

for $z$ result in no further improvements and the best performance is achieved for $z = 4$. As compared to causal dynamic attention, causal local attention showed better results. The observations verified the inference that extensive previous information is not required in speech enhancement. This inference is logical since noisy situations, both types and SNRs, change over time. The observations are valid for the attention networks since the attention LSTM outperformed the baseline LSTM.

**Table 4. SDR in matched test scores.**

| Noise | Algorithm | -5dB | 0dB | 5dB | Avg |
|---|---|---|---|---|---|
| Airport Noise | Noisy (UnP) | -4.78 | 0.11 | 5.07 | 0.13 |
| | DNN-IRM | 3.98 | 6.86 | 8.38 | 6.41 |
| | LSTM-IRM | 4.09 | 7.10 | 9.54 | 6.91 |
| | LSTM-KF | 4.05 | 7.13 | 9.55 | 6.92 |
| | FDNN-KF | 3.99 | 7.01 | 9.42 | 6.80 |
| | DeepXi-KF | 4.09 | 7.15 | 9.84 | 7.02 |
| | DeepResGRU-KF | 4.16 | 7.22 | 10.1 | 7.16 |
| | Proposed | 4.21 | 7.33 | 10.7 | 7.41 |
| Babble Noise | Noisy (UnP) | -4.73 | 0.13 | 5.08 | 0.16 |
| | DNN-IRM | 3.82 | 6.31 | 8.74 | 6.29 |
| | LSTM-IRM | 3.95 | 6.58 | 9.05 | 6.52 |
| | LSTM-KF | 4.03 | 7.21 | 9.25 | 6.83 |
| | FDNN-KF | 3.95 | 7.10 | 9.12 | 6.72 |
| | DeepXi-KF | 4.11 | 7.26 | 9.24 | 6.87 |
| | DeepResGRU-KF | 4.19 | 7.31 | 9.29 | 6.93 |
| | Proposed | 4.28 | 7.40 | 9.36 | 7.01 |
| Car Noise | Noisy (UnP) | -4.85 | 0.08 | 5.05 | 0.09 |
| | DNN-IRM | 3.81 | 6.42 | 8.83 | 6.35 |
| | LSTM-IRM | 4.20 | 7.21 | 9.92 | 7.11 |
| | LSTM-KF | 4.28 | 7.29 | 10.1 | 7.20 |
| | FDNN-KF | 3.92 | 6.52 | 9.01 | 6.46 |
| | DeepXi-KF | 4.34 | 7.35 | 10.2 | 7.45 |
| | DeepResGRU-KF | 4.48 | 7.44 | 10.4 | 7.58 |
| | Proposed | 4.56 | 7.59 | 10.8 | 7.65 |
| Factory Noise | Noisy (UnP) | -4.69 | 0.12 | 5.07 | 0.17 |
| | DNN-IRM | 3.66 | 5.34 | 8.72 | 6.24 |
| | LSTM-IRM | 3.85 | 5.53 | 9.52 | 6.30 |
| | LSTM-KF | 3.83 | 5.78 | 9.63 | 6.41 |
| | FDNN-KF | 3.75 | 5.55 | 8.87 | 6.05 |
| | DeepXi-KF | 3.91 | 6.12 | 9.96 | 6.66 |
| | DeepResGRU-KF | 3.89 | 6.25 | 10.2 | 6.78 |
| | Proposed | 4.01 | 6.69 | 10.3 | 6.99 |
| Street Noise | Noisy (UnP) | -4.76 | 0.11 | 4.99 | 0.11 |
| | DNN-IRM | 4.01 | 6.83 | 8.34 | 6.39 |
| | LSTM-IRM | 4.05 | 7.32 | 9.22 | 6.86 |
| | LSTM-KF | 4.08 | 7.43 | 9.33 | 6.94 |
| | FDNN-KF | 4.02 | 6.98 | 8.40 | 6.46 |
| | DeepXi-KF | 4.11 | 7.86 | 9.66 | 7.21 |
| | DeepResGRU-KF | 4.18 | 7.73 | 9.89 | 7.26 |
| | Proposed | 4.19 | 7.82 | 10.1 | 7.37 |

Table 8 shows the comparison between the loss errors and predicted results of the DWAtten-LSTM with and without the DW loss function. The DW loss function improved the predicted scores (STOI and PESQ). The errors are reduced by weighted MSE ($3.42 \times 10^{-4}$) as compared to a non-weighted MSE ($3.54 \times 10^{-4}$).

To understand the attention process, the attention maps are illustrated in Fig 5. The x-axis denotes $h^K$ and the y-axis denotes $h^Q$. The points, $(x;y)$ denote the attention weights. The

**Table 5. Unmatched test scores in five noise sources at all SNRs.** The results are averaged over all testing utterances.

| Noise | Algorithm | STOI (in%) | PESQ | SDR |
|-------|-----------|-----------|------|-----|
| Airport Noise | Noisy (UnP) | 71.6 | 1.83 | 0.14 |
| | DNN-IRM | 82.6 | 2.27 | 6.30 |
| | LSTM-IRM | 85.0 | 2.37 | 6.84 |
| | LSTM-KF | 85.9 | 2.40 | 6.96 |
| | FDNN-KF | 83.4 | 2.31 | 6.41 |
| | DeepXi-KF | 86.7 | 2.45 | 7.21 |
| | DeepResGRU-KF | 86.9 | 2.46 | 7.31 |
| | Proposed | 87.6 | 2.48 | 7.39 |
| Babble Noise | Noisy (UnP) | 65.9 | 1.72 | 0.19 |
| | DNN-IRM | 76.7 | 2.12 | 6.13 |
| | LSTM-IRM | 78.5 | 2.23 | 6.31 |
| | LSTM-KF | 79.1 | 2.28 | 6.35 |
| | FDNN-KF | 77.2 | 2.15 | 6.18 |
| | DeepXi-KF | 80.2 | 2.30 | 6.45 |
| | DeepResGRU-KF | 80.8 | 2.32 | 6.51 |
| | Proposed | 81.0 | 2.35 | 6.62 |
| Car Noise | Noisy (UnP) | 67.5 | 1.65 | 0.13 |
| | DNN-IRM | 79.6 | 2.17 | 6.38 |
| | LSTM-IRM | 82.8 | 2.33 | 6.92 |
| | LSTM-KF | 83.1 | 2.40 | 7.01 |
| | FDNN-KF | 80.1 | 2.21 | 6.45 |
| | DeepXi-KF | 84.2 | 2.45 | 7.12 |
| | DeepResGRU-KF | 84.8 | 2.48 | 7.28 |
| | Proposed | 86.0 | 2.49 | 7.45 |
| Factory Noise | Noisy (UnP) | 66.1 | 1.61 | 0.18 |
| | DNN-IRM | 75.7 | 2.08 | 5.92 |
| | LSTM-IRM | 78.6 | 2.24 | 6.39 |
| | LSTM-KF | 78.8 | 2.29 | 6.45 |
| | FDNN-KF | 76.0 | 2.10 | 8.98 |
| | DeepXi-KF | 79.5 | 2.35 | 6.57 |
| | DeepResGRU-KF | 80.2 | 2.38 | 6.71 |
| | Proposed | 81.3 | 2.39 | 6.87 |
| Street Noise | Noisy (UnP) | 66.8 | 1.73 | 0.22 |
| | DNN-IRM | 77.8 | 2.18 | 6.06 |
| | LSTM-IRM | 81.5 | 2.31 | 6.34 |
| | LSTM-KF | 81.9 | 2.35 | 6.40 |
| | FDNN-KF | 78.0 | 2.21 | 6.10 |
| | DeepXi-KF | 82.6 | 2.38 | 6.45 |
| | DeepResGRU-KF | 84.7 | 2.40 | 6.58 |
| | Proposed | 85.1 | 2.42 | 6.63 |

attention-based network assigns different attention levels (weights) to the contextual frames. The top spectrogram shows noisy speech, and the other spectrogram shows clean speech, respectively.

In experiments, time-varying spectral analysis is conducted to showcase the performance of DWAtten-LSTM. Fig 6 demonstrates the sample spectrogram analysis. A clean speech

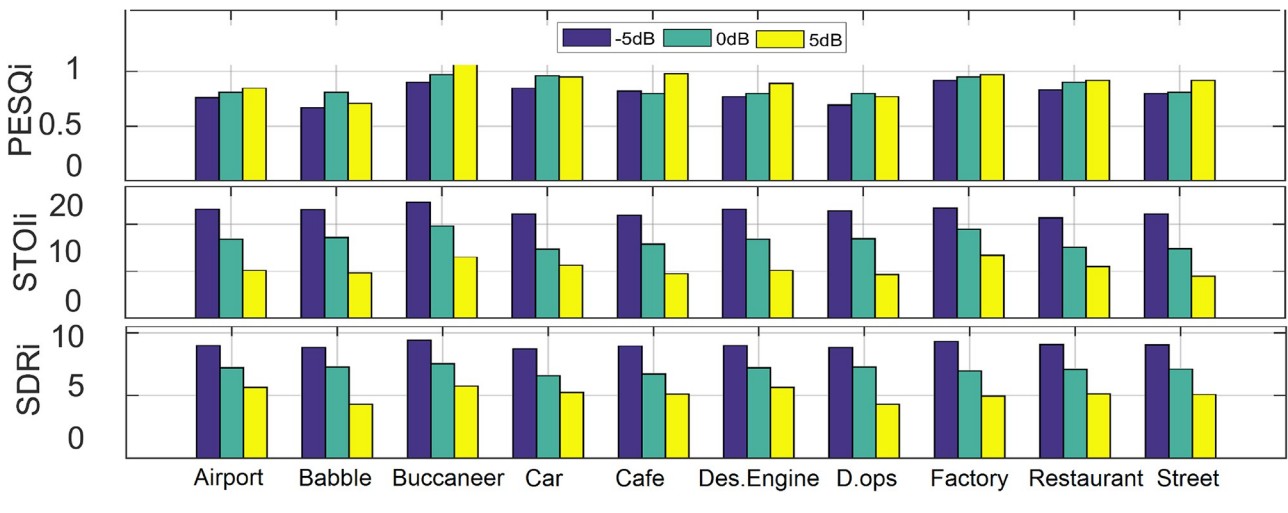

**Fig 3.**

utterance is mixed with babble noise at 5 dB. The spectrogram of DWAtten-LSTM is plotted in Fig 6(F). The harmonic structures of the vowel and the formant peaks are well retained. Moreover, the spectrogram showed excellent structure during speech activity. During the speech pause, DWAtten-LSTM removed the residual noise signals. The weak harmonic structures in the high-frequency sub-bands are well maintained. Thus, a better speech quality of the enhanced speech is achieved by DWAtten-LSTM. The weak energy in the speech utterance is also well retained and yields less speech distortion. Therefore, the intelligibility of noisy speech is improved. The residual noise signals are evident in the spectrograms of LMMSE and OM-LSA, plotted in Fig 6(C) and 6(D).

The complexity and convergence analysis are also given. The complexity of a deep learning algorithm revolves around the number of training parameters; LSTM networks have 1.2 million parameters. This is clearly a fewer number as compared to other networks used for speech enhancement, for example, 10 million parameters are used by the residual LSTM [47]. This also significantly reduces the training time and speeds up the process. DWAtten-LSTM took less time per epoch compared to the Residual LSTM (using an NVIDIA GTX 950 Ti GPU). Next, we observed the convergence of Weighted-MSE between the estimated and true values

**Table 6. Comparison against non-deep learning methods.** Test Scores are averaged over five noise sources at all SNRs.

|  | STOI (in %) |  |  |  | PESQ |  |  |  | SDR |  |  |  |
| --- | --- | --- | --- | --- | --- | --- | --- | --- | --- | --- | --- | --- |
| Algorithm | -5dB | 0dB | 5dB | Avg | -5dB | 0dB | 5dB | Avg | -5dB | 0dB | 5dB | Avg |
| Noisy (UnP) | 58.02 | 68.87 | 80.05 | 68.98 | 1.43 | 1.73 | 2.01 | 1.72 | -4.73 | 0.13 | 5.07 | 0.13 |
| DNN-IRM | 73.94 | 80.06 | 85.39 | 79.79 | 1.83 | 2.23 | 2.61 | 2.22 | 3.84 | 6.40 | 8.50 | 6.25 |
| LSTM-IRM | 77.39 | 83.15 | 87.89 | 82.81 | 2.02 | 2.36 | 2.72 | 2.37 | 4.00 | 6.61 | 9.35 | 6.65 |
| LLMSE | 59.21 | 70.22 | 81.80 | 70.41 | 1.49 | 1.75 | 2.22 | 1.82 | -3.43 | 0.18 | 5.32 | 0.69 |
| OM-LSA | 59.49 | 71.50 | 82.14 | 71.04 | 1.52 | 1.81 | 2.20 | 1.84 | -3.88 | 0.21 | 5.51 | 0.61 |
| Proposed-M | 82.46 | 87.20 | 91.84 | 87.16 | 2.24 | 2.62 | 2.92 | 2.59 | 4.19 | 7.17 | 10.3 | 7.20 |
| Proposed-UM | 79.06 | 84.08 | 89.48 | 84.20 | 2.09 | 2.40 | 2.79 | 2.43 | 4.09 | 6.72 | 10.0 | 6.97 |
| Proposed-Avg | 80.76 | 85.64 | 90.66 | 85.68 | 2.17 | 2.51 | 2.85 | 2.51 | 4.14 | 6.95 | 10.2 | 7.09 |

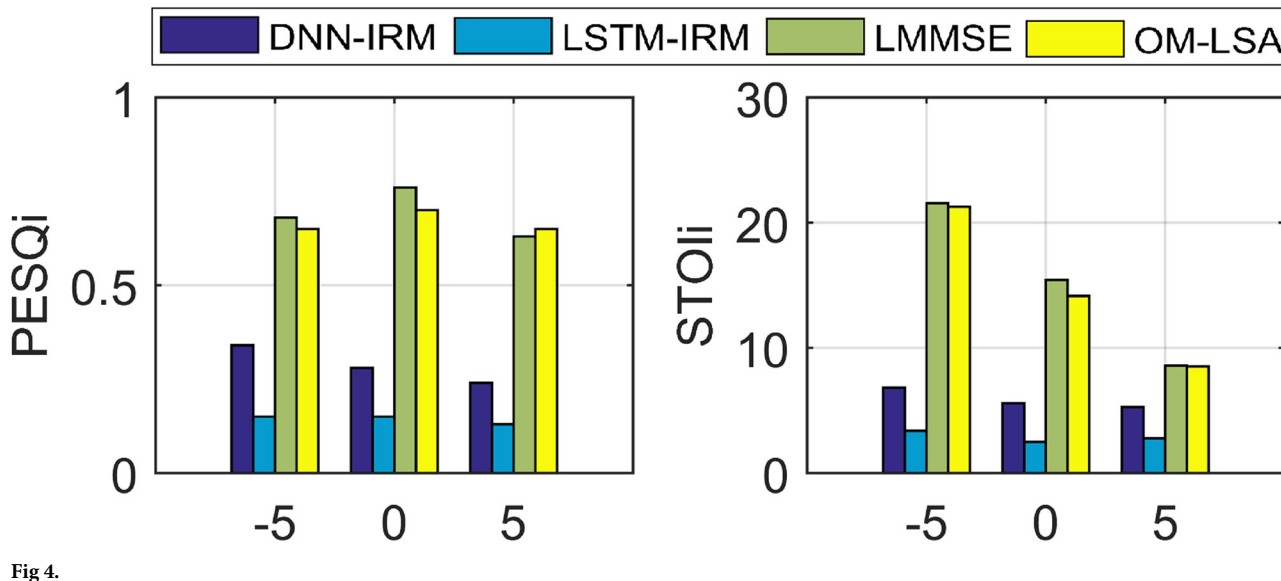

**Fig 4.**

for the training and testing data sets of DWAtten-LSTM. The MSE has been reduced after every epoch until converging at around epoch 155.

According to STOI, PESQ, and SDR, the following inferences are drawn. Under various noisy situations, PESQ, STOI, and SDR values indicate that DWAtten-LSTM achieved the best improvements in quality (PESQi), intelligibility (STOIi), and speech distortion (SDRi) as compared to the competing deep learning and non-deep learning methods. The proposed DWAtten-LSTM method improved the quality without degrading speech intelligibility in noisy situations. All deep-learning methods showed repeated improvements in STOI and SDR values, which suggests the potential of deep learning for speech enhancement tasks.

The ASR systems use a magnitude spectrum of speech signals, and one would expect that deep learning approaches would certainly improve ASR performance in noisy situations. For ASR systems, SE algorithms operate at the front end. We have used Google ASR [48] to examine the ASR performance in terms of the WERs. The average WERs are given in Table 9,

**Table 7. Unmatched test scores in five noise sources at all SNRs.** The results are averaged over all testing utterances.

| z | STOI (in%) | | | PESQ | | |
|---|---|---|---|---|---|---|
| | -5dB | 0dB | 5dB | -5dB | 0dB | 5dB |
| 4 | 80.92 | 86.46 | 90.88 | 2.163 | 2.503 | 2.81 |
| 12 | 80.90 | 86.42 | 90.82 | 2.160 | 2.501 | 2.80 |
| 24 | 80.70 | 86.12 | 90.01 | 2.145 | 2.489 | 2.77 |

**Table 8. Dynamical-Weight vs. Non-Dynamical-Weight loss.**

| Algorithm | Objective Measure | Errors |
|---|---|---|
| Proposed+MSE | STOI: 83%, PESQ: 2.42 | $3.54 \times 10^{-4}$ |
| Proposed+DW-MSE | STOI: 85%, PESQ: 2.50 | $3.42 \times 10^{-4}$ |
| Improvements | STOIi: 2.1%, PESQi: 3.3% | -3.51% |

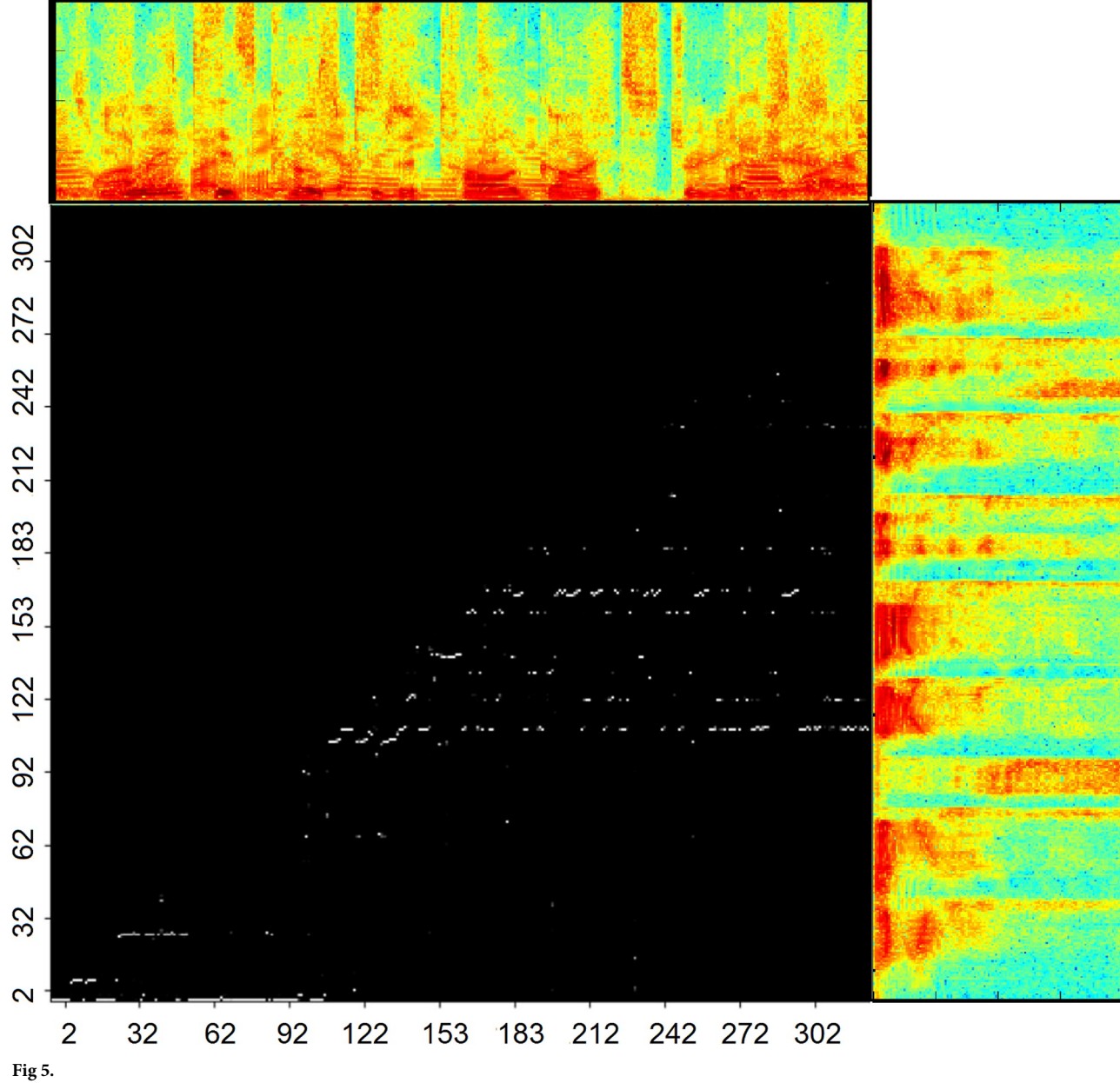

**Fig 5.**

which shows that LSTM and DWAtten-LSTM boosted the ASR performance. The error rates decreased from 46.33% (noisy signals) to 13.11% (DWAtten-LSTM) and 15.73% (LSTM), respectively. The ASR gradually decreases as the SNR increases, partly because the noise becomes smaller. The ASR experiments aim to show the potential of the proposed RNNs and DNNs instead of achieving state-of-the-art (SOTA) results.

## 4.1 Subjective evaluation

In addition, we have conducted subjective listening tests to assess the perceptual quality of enhanced speech. The enhanced speech utterances are randomly chosen from various noise

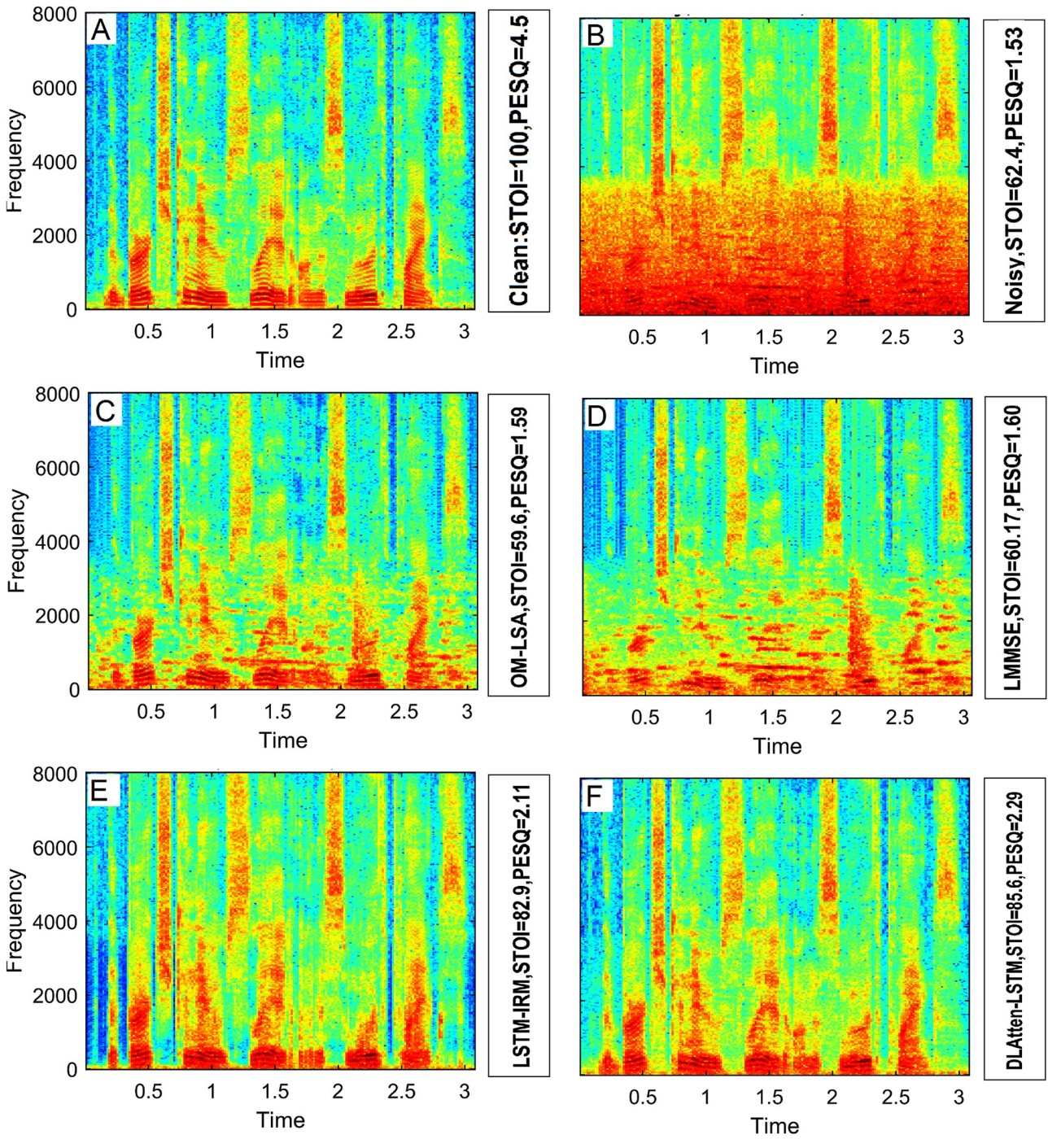

**Fig 6.**

**Table 9. WERs for different SE algorithms.**

| Noisy Speech | DWAtten-LSTM | LSTM | LSTM-KF | LMMSE |
|---|---|---|---|---|
| 46.33% | 13.11% | 15.73% | 14.97% | 27.68% |

**Table 10. The subjective listener's biodata.**

| Listeners | L1 | L2 | L3 | L4 | L5 | L6 | L7 | L8 | L9 |
|---|---|---|---|---|---|---|---|---|---|
| Age | 27 | 29 | 33 | 36 | 40 | 40 | 45 | 48 | 48 |
| Gender | M | M | F | M | F | F | M | M | M |

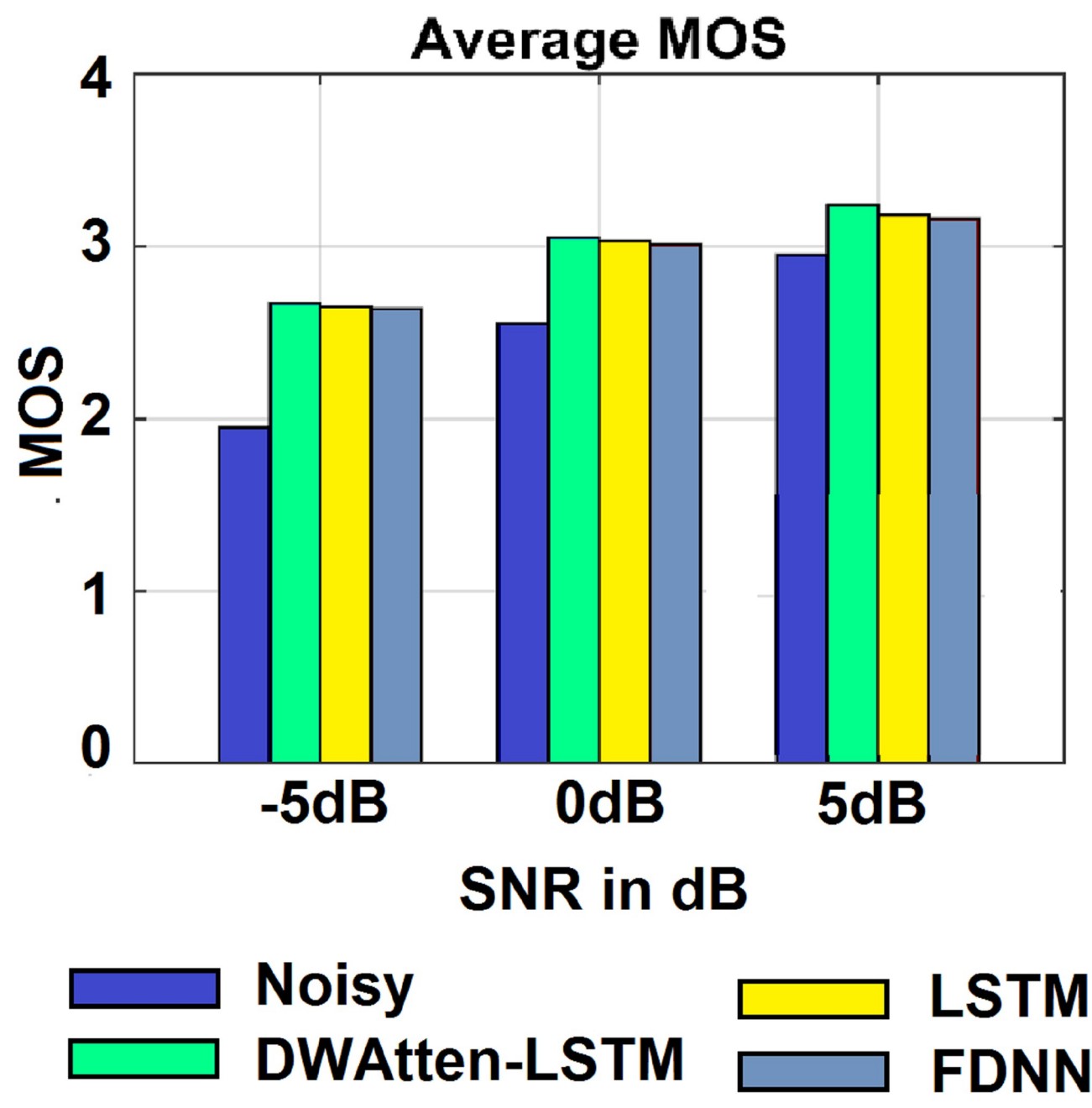

**Fig 7.**

**Table 11. Performance analysis of proposed SE in reverberant situations.**

| Metric | Method | Reverberation Time | | |
|--------|--------|:--------:|:--------:|:--------:|
| | | **0.4 sec** | **0.6 sec** | **0.8 sec** |
| STOI | Noisy Reverb | 53.1 | 48.5 | 40.2 |
| | Wu and Wang [49] | 65.3 | 59.8 | 55.1 |
| | Proposed | 78.3 | 70.1 | 68.4 |
| PESQ | Noisy Reverb | 2.11 | 1.98 | 1.81 |
| | Wu and Wang [49] | 2.32 | 2.00 | 1.92 |
| | Proposed | 2.45 | 2.19 | 2.04 |

sources (airport, babble, factory, and restaurant) using three SNRs, which are -5 dB, 0 dB, and 5 dB. In total, 300 speech utterances are used to assess DNN, LSTM, and the proposed SE. The participants are requested to assign a score (from 0 to 5) according to perceived speech quality. During experiments, no speech utterance is repeated. The listening tests are conducted in an isolated room using high-quality headphones. The data of the listeners who participated in the subjective listening tests for speech quality are given in Table 10. Prior training sessions are arranged to educate the listeners about the procedures.

Fig 7 shows the subjective listening tests in terms of MOS. The proposed DWAtten-LSTM showed better MOS performance. The average MOS result at negative SNRs is higher than 2.80 (MOS≥2.86 at -5 dB), which indicates significant performance. At SNR≥0dB, the DWAtten-LSTM exceeded the average MOS score beyond 3.0 (MOS≥3.0 at SNR≥0dB). The MOS results for all listeners in the tests are averaged. The ANOVA tests for MOS at -5 dB, 0 dB, and 5 dB are [$F_{(2, 9)}$ = 43.4, $p < 0.0001$], [$F_{(2, 9)}$ = 34.7, $p < 0.0001$] and [$F_{(2, 9)}$ = 28.3, $p < 0.0001$] which indicate the statistical significance achieved by DWAtten-LSTM in terms of the MOS scores. The other models (DNN and LSTM) also performed better since deep learning is able to produce better speech quality.

## 4.2 Speech dereverberation

This section examined the dereverberation performance of the proposed SE. To train the SE model, three reverberation times (0.4 sec, 0.6 sec, and 0.8 sec) are considered. A total of 100 anechoic speech utterances from the IEEE dataset [40] are used to create the training dataset. The testing dataset contains 40 reverberant speech utterances. There is no overlap between the speech utterances used during model training and testing. The proposed method with reverberant speech utterances is compared and examined for dereverberation. The results are compared with the study of Wu and Wang [49], where estimated inverse filters and spectral subtraction are used to reduce reverberation. Table 11 shows the results using STOI and PESQ. The proposed method delivered the best STOI and PESQ scores, i.e., STOI≥78.3%, and PESQ≥2.45 at RT≥4 sec. The spectrograms are provided in Fig 8, where the smearing energy produced by reverberation is considerably reduced, showing that the reverberation performance of the proposed method is improved.

## 5 Conclusions

In this paper, we have proposed a monaural SE based on the attention LSTM encoder-decoder model with a novel loss function. The proposed DWAtten-LSTM estimated the magnitude spectrum from the noisy speech signals using an ideal ratio mask. We have compared this model to the baseline and competing for deep learning and non-deep-learning methods for

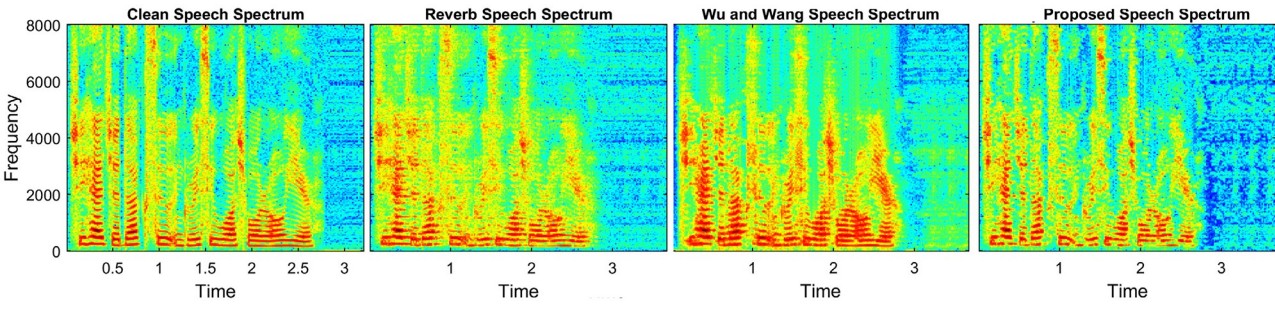

**Fig 8.**

speech intelligibility and quality assessment. The objective assessments are accomplished in various noisy situations using three input SNR levels. The PESQ and SDR values indicated that the proposed DWAtten-LSTM achieved significant gains of 0.79 (45.93%) and 6.96dB over noisy speech. Similarly, STOI indicated that DWAtten-LSTM kept intelligibility in all noisy situations and STOI achieved a large gain of 16.70% over the noisy speech. The subjective analysis confirmed the success of the proposed model in terms of speech quality. The results and analysis concluded that we achieved better results in terms of speech quality and intelligibility with the proposed DWAtten-LSTM. The attention process observations verified the inference that extensive previous information is not vital in speech enhancement. The proposed loss function significantly improved the model learning. Although deep learning for speech enhancement outperformed the conventional methods with their complex network architectures, yet required less computationally complex and efficient network architectures for improved performance. The proposed DWAtten-LSTM SE algorithm has demonstrated considerable performance gain as compared to the baseline LSTM and FDNN and achieved higher performance gains when compared to the conventional SE.

Our future research will focus on further improving the quality and intelligibility by proposing computationally less complex network architectures in intense unseen noises and speakers. Moreover, phase estimation will also be included to increase the speech quality. This study used STFT as a transformation tool for frequency domain representation; however, several transformations are available in the literature. In future studies, these transformations [50–53] will be used for more in-depth analysis.

## Supporting information

**S1 File.**
(ZIP)

**S2 File.**
(ZIP)

**S3 File.**
(ZIP)

## Author Contributions

**Conceptualization:** Fahad Khalil Peracha, Muhammad Irfan Khattak.

**Data curation:** Fahad Khalil Peracha.

**Formal analysis:** Muhammad Irfan Khattak.

**Investigation:** Muhammad Irfan Khattak, Nasir Saleem.

**Methodology:** Fahad Khalil Peracha.

**Resources:** Nema Salem.

**Software:** Fahad Khalil Peracha, Nasir Saleem.

**Supervision:** Nasir Saleem.

**Validation:** Nema Salem.

**Writing – original draft:** Fahad Khalil Peracha.

**Writing – review & editing:** Nasir Saleem.

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
