## [Decision Letter · Decision Letter 0]

26 Dec 2022

PONE-D-22-32395Causal Speech Enhancement using Dynamical-Weighted Loss and Attention Encoder-Decoder Recurrent Neural NetworkPLOS ONE

Dear Dr. Saleem,

Thank you for submitting your manuscript to PLOS ONE. After careful consideration, we feel that it has merit but does not fully meet PLOS ONE’s publication criteria as it currently stands. Therefore, we invite you to submit a revised version of the manuscript that addresses the points raised during the review process.

We look forward to receiving your revised manuscript.

Kind regards,

Abdullah M. Mutawa, Ph.D

Academic Editor

PLOS ONE

2. Please note that PLOS ONE has specific guidelines on code sharing for submissions in which author-generated code underpins the findings in the manuscript. In these cases, all author-generated code must be made available without restrictions upon publication of the work. Please review our guidelines at https://journals.plos.org/plosone/s/materials-and-software-sharing#loc-sharing-code and ensure that your code is shared in a way that follows best practice and facilitates reproducibility and reuse. New software must comply with the Open Source Definition.

4. Please upload a new copy of Figure 1 as the detail is not clear. Please follow the link for more information: " ext-link-type="uri" xlink:type="simple">https://blogs.plos.org/plos/2019/06/looking-good-tips-for-creating-your-plos-figures-graphics/"
https://blogs.plos.org/plos/2019/06/looking-good-tips-for-creating-your-plos-figures-graphics/

Additional Editor Comments:

Dear Author,

Please answer all points mentioned by the reviewers and prepare a point-by-point table with reply to every point showing the location of change in the manuscript.

Reviewers' comments:

Reviewer's Responses to Questions

**Comments to the Author**

1. Is the manuscript technically sound, and do the data support the conclusions?

Reviewer #1: Partly

Reviewer #2: Yes

2. Has the statistical analysis been performed appropriately and rigorously? 

Reviewer #1: No

Reviewer #2: N/A

3. Have the authors made all data underlying the findings in their manuscript fully available?

Reviewer #1: No

Reviewer #2: Yes

4. Is the manuscript presented in an intelligible fashion and written in standard English?

Reviewer #1: Yes

Reviewer #2: No

5. Review Comments to the Author

Reviewer #1: Causal Speech Enhancement using Dynamical-Weighted Loss and Attention EncoderDecoder Recurrent Neural Network" is a good topic for paper but having some suggestions

1) Compare results with some current developed methods and use those references in your introduction.

2) use one more different input data for analysis and evaluation.

3) use proper references

Reviewer #2: Summary:

In this work, a causal data-driven model is proposed for single-microphone SE operating in real-time systems. The proposed system utilizes attention encoder-decoder long short-term memory (LSTM) to estimate the time-frequency mask from noisy speech.

The manuscript is interesting; however, the following comment should be addressed :

Abstract :

- - - - - - - - - - -

1 – Please include problem statement .

2 - Improvement ratio between the proposed and existing works should be included .

Introduction Section :

- - - - - - - - - - - - - - - - - - - - - -

3 – In the Introduction, the authors need to refer to other speech enhancement algorithms such as : i) doi: 10.1088/1757-899X/1090/1/012102, ii) doi: 10.3390/s21217025, and ii) doi: 10.1109/ACCESS.2019.2929864.

4 – The contribution should be included as a list for better readability.

Proposed Speech Enhancement Section :

- - - - - - - - - - - - - - - - - - - - - - - - - - - - - - - - - - -

5 – Please check the numbering of the subsections .

6 – Define the functions used such as “|⋅|” , “||⋅||”, “*” , etc .

Experiments Section :

- - - - - - - - - - - - - - - - - - - - - -

7 – The authors utilize STFT; however, there are different types of transforms which are based on orthogonal polynomials. The authors need to refer to the difference between the Fourier Transform and the following transforms Krawtchouk transform (doi: 10.3390/e23091162), Hahn Transform (doi: 10.1109/ACCESS.2022.3170893), and Meixner transform (doi: 10.1007/s11554-021-01093-z). This will help the researchers to utilize other transforms for SE.

Conclusion Section :

- - - - - - - - - - - - - - - - - - - - - -

8 – This section is fine. No comments .

General Comments:

- - - - - - - - - - - - - - - - -

9 - There are some grammatical error should be checked and corrected .

- - - - - - - - - - - - - - - - - - - - - - - - - - - - - - - - - - - - - - - - - - - - - - - - - - - - - - - - - - - - - - - - - - - - - - - - - - - - - - - - - - - - - - - - - - - - - - - - - - - - - - - - - - - - - - - - - - - - - - - - - - - - - - - - - - - - - - - - - - - - - - - - - - - - - - - - - - - - - - - - - - - - - - - - - - - - - - - - - - - - - - - - - - - - - - - - - - - - - - - - - - - - - - - - - - - - - - - - - - - - - - - - - - - - - - - - - - - - - - - - - - - - - - - - - - - - - - - - - - - - - - - - - - - - - - - - - - - - - - - - - - - - - - - - - - - - - - - - - - - - - - - - - - - - - - - - - - - - - - - - - - - - - - - - - - - - - - - - - - - - - - - - - - - - - - - -

6. PLOS authors have the option to publish the peer review history of their article (what does this mean?). If published, this will include your full peer review and any attached files.

Reviewer #1: No

Reviewer #2: No

quillbot-extension-portal/quillbot-extension-portal

---

## [Author Response · Author response to Decision Letter 0]

23 Jan 2023

Reviewer #1: Causal Speech Enhancement using Dynamical-Weighted Loss and Attention Encoder Decoder Recurrent Neural Network" is a good topic for paper but having some suggestions

1- Compare results with some current developed methods and use those references in your introduction.

Response: Thank you for the important suggestion, the results are compared to the recently developed methods and the reference methods are reflected in the introduction part of the revised manuscript. The Tables are revised to address the suggestion of respected reviewer. Table captions are highlighted with Red to show the changes. 

2- Use one more different input data for analysis and evaluation.

Response: Thank you for the important suggestion. This algorithm was intended for the additive noisy backgrounds; however, to address the concern of the respected reviewer, the authors have included noisy reverberation as other input data. 

Action: The following text and results are added to the revised paper.

This section examined the dereverberation performance of the proposed SE. To train the SE model, three reverberation times (0.4 sec, 0.6 sec, and 0.8 sec) are considered. A total of 100 anechoic speech utterances from the IEEE dataset [40] are used to create the training dataset. The testing dataset contains 40 reverberant speech utterances. There is no overlapping between the speech utterances used during model training, and testing. The proposed method with reverberant speech utterances are compared and examined for the dereverberation. The results are compared with study of Wu and Wang [49], where estimated inverse filters and spectral subtraction are used to reduce the reverberation. Table 11 shows the results using STOI and PESQ. The proposed method delivered the best STOI and PESQ scores, i.e., STOI≥78.3%, and PESQ≥2.45 at RT≥4 sec. The spectrograms are provided in Fig. 8 where the smearing energy produced by reverberation is considerably reduced, showing that the reverberation performance of the proposed method. 

3- Use proper references.

Response: Thank you for the suggestion, the references is arranged in the proper manner to address the concern

Reviewer#2: The manuscript is interesting; however, the following comment should be addressed

1 – Please include problem statement.

Response: Thank you for the important suggestion, the problem statement is included in the revised manuscript. 

Action: Speech enhancement (SE) reduces background noise signals in target speech and is applied at the front end in various real-world applications, including robust ASRs and real-time processing in mobile phone communications. SE systems are commonly integrated into mobile phones to increase quality and intelligibility. As a result, a low-latency system is required to operate in real-world applications. On the other hand, these systems need efficient optimization. This research focuses on the single-microphone SE operating in real-time systems with better optimization.

2 - Improvement ratio between the proposed and existing works should be included.

Response: Thank you for the suggestion, the improvement ratios between the proposed and related studies are included in the revised manuscript. 

3 – In the Introduction, the authors need to refer to other speech enhancement algorithms such as : i) doi: 10.1088/1757-899X/1090/1/012102, ii) doi: 10.3390/s21217025, and ii) doi: 10.1109/ACCESS.2019.2929864.

Response: The introduction part is modified with the references suggested. Thank you 

4 – The contribution should be included as a list for better readability.

Response: The contributions are listed in the revised manuscript to address the concern of the reviewer. Thank you 

5 – Please check the numbering of the subsections.

Response: Thank you for the correction, the sections and subsections are corrected in the revised manuscript. 

6 – Define the functions used such as “|⋅|” , “||⋅||”, “*” , etc .

Response: Thank you for the important correction, the typos in equations is corrected and these notations are defined in the revised manuscript.

7 – The authors utilize STFT; however, there are different types of transforms which are based on orthogonal polynomials. The authors need to refer to the difference between the Fourier Transform and the following transforms Krawtchouk transform (doi: 10.3390/e23091162), Hahn Transform (doi: 10.1109/ACCESS.2022.3170893), and Meixner transform (doi: 10.1007/s11554-021-01093-z). This will help the researchers to utilize other transforms for SE.

Response: Thank you for the important suggestion, the authors have used STFT transform widely used in the speech signal processing. The other mentioned transforms are associated to the different applications and the authors will conduct a separate study based on these transforms. The mentioned transforms are added with the references to the revised manuscript.

---

## [Decision Letter · Decision Letter 1]

27 Apr 2023

Causal Speech Enhancement using Dynamical-Weighted Loss and Attention Encoder-Decoder Recurrent Neural Network

PONE-D-22-32395R1

Dear Dr. Saleem,

We’re pleased to inform you that your manuscript has been judged scientifically suitable for publication and will be formally accepted for publication once it meets all outstanding technical requirements.

Kind regards,

Abdullah M. Mutawa, Ph.D

Academic Editor

PLOS ONE

Reviewers' comments:

Reviewer's Responses to Questions

**Comments to the Author**

Reviewer #2: All comments have been addressed

Reviewer #3: All comments have been addressed

2. Is the manuscript technically sound, and do the data support the conclusions?

Reviewer #2: Yes

Reviewer #3: Yes

3. Has the statistical analysis been performed appropriately and rigorously? 

Reviewer #2: Yes

Reviewer #3: Yes

4. Have the authors made all data underlying the findings in their manuscript fully available?

Reviewer #2: Yes

Reviewer #3: Yes

5. Is the manuscript presented in an intelligible fashion and written in standard English?

Reviewer #2: Yes

Reviewer #3: Yes

6. Review Comments to the Author

Reviewer #2: Summary:

In this work, a causal data-driven model is proposed for single-microphone SE operating in real-time systems. The proposed system utilizes attention encoder-decoder long short-term memory (LSTM) to estimate the time-frequency mask from noisy speech.

The authors have addressed the raised comments. No further comments.

Comments:

Abstract :

- - - - - - - - - - -

1 – The abstract is fine. No further comments.

Introduction Section :

- - - - - - - - - - - - - - - - - - - - - -

2 – This section is fine. No further comments.

Proposed Speech Enhancement Section :

- - - - - - - - - - - - - - - - - - - - - - - - - - - - - - - - - - -

3 – This section is fine. No further comments.

Experiments Section :

- - - - - - - - - - - - - - - - - - - - - -

4 – This section is fine. No further comments.

Conclusion Section :

- - - - - - - - - - - - - - - - - - - - - -

5 – This section is fine. No further comments.

- - - - - - - - - - - - - - - - - - - - - - - - - - - - - - - - - - - - - - - - - - - - - - - - - - - - - - - - - - - - - - - - - - - - - - - - - - - - - - - - - - - - - - - - - - - - - - - - - - - - - - - - - - - - - - - - - - - - - - - - - - - - - - - - - - - - - - - - - - - - - - - - - - - - - - - - - - - - - - - - - - - - - - - - - - - - - - - - - - - - - - - - - - - - - - - - - - - - - - - - - - - - - - - - - - - - - - - - - - - - - - - - - - - - - - - - - - - - - - - - - - - - - - - - - - - - - - - - - - - - - - - - - - - - - - - - - - - - - - - - - - - - - - - - - - - - - - - - - - - - - - - - - - - - - - - - - - - - - - - - - - - - - - - - - - - - - - - - - - - - - - - - - - - - - - - - - - - - - - - - - - - - - - - - - - - - - - - - - - - - - - - - - - - - - - - - - - - - - - - - - - - - - - - - - - - - - - - - - - - - - - - - - - - - - - - - - - - - - - - - - - - - - - - - - - - - - - - - - - - - - - - - - - - - - - - - - - - - - - - - - - - - - - - - - - - - - - - - - - - - - - - - - - - - - - - - - - - - - - - - - - - - - - - - - - - - - - - - - - - - - - - - - - - - - - - - - - - - - - - - - - - - - - - - - - - - - - - - - - - - - - - - - - - - - - - - - - - - - - - - - - - - - - - - - - - - - - - - - - - - - - - - - - - - - - - - - - - - - - - - - - - - - - - - - - - - - - - - - - - - - - - - - - - - - - - - - - - - - - - - - - - - - - - - - - - - - - - - - - - - - - - -

Reviewer #3: I noticed that the revised version corrects all recommendations form the reviewers, I accept the paper in this form.

7. PLOS authors have the option to publish the peer review history of their article (what does this mean?). If published, this will include your full peer review and any attached files.

Reviewer #2: No

Reviewer #3: No

---

## [Editor Report · Acceptance letter]

2 May 2023

PONE-D-22-32395R1 

Causal Speech Enhancement using Dynamical-Weighted Loss and Attention Encoder-Decoder Recurrent Neural Network 

Dear Dr. Saleem:

I'm pleased to inform you that your manuscript has been deemed suitable for publication in PLOS ONE. Congratulations! Your manuscript is now with our production department. 

Kind regards, 

on behalf of

Dr. Abdullah M. Mutawa 

Academic Editor

PLOS ONE